# Effect of an *Alliaceae* Encapsulated Extract on Growth Performance, Gut Health, and Intestinal Microbiota in Broiler Chickens Challenged with *Eimeria* spp.

**DOI:** 10.3390/ani13243884

**Published:** 2023-12-18

**Authors:** Gonzalo Villar-Patiño, María del Carmen Camacho-Rea, Myrna Elena Olvera-García, Julio César Baltazar-Vázquez, Gabriela Gómez-Verduzco, Guillermo Téllez, Aurora Labastida, Aurora Hilda Ramírez-Pérez

**Affiliations:** 1Programa de Doctorado en Ciencias de la Salud y de la Producción Animal, Facultad de Medicina Veterinaria y Zootecnia, Universidad Nacional Autónoma de México, Avenida Universidad 3000, Coyoacán, Ciudad de Mexico 04510, Mexico; gvillar@gponutec.com; 2Grupo Nutec, Avenida de las Fuentes No. 14, Parque Industrial Bernardo Quintana, El Marqués 76246, Querétaro, Mexico; molvera@gponutec.com; 3Instituto Nacional de Ciencias Médicas y Nutrición “Salvador Zubirán”, Departamento de Nutrición Animal, Tlalpan, Ciudad de Mexico 14080, Mexico; 4Specialized Animal Nutrition Research Network, Grupo Nutec, La Valla, San Juan del Río 76814, Querétaro, Mexico; jbaltazar@gponutec.com; 5Facultad de Medicina Veterinaria y Zootecnia, Universidad Nacional Autónoma de México, Avenida Universidad 3000, Coyoacán, Ciudad de Mexico 04510, Mexico; gagove@unam.mx; 6Department of Poultry Science, University of Arkansas, Fayetteville, AR 72701, USA; gtellez@uark.edu; 7OMICs Analysis, Camino a Xilotepetl No. 45, Tepoztlán 62520, Morelos, Mexico; a.labastida@omicsanalysis.mx

**Keywords:** broiler chickens, coccidiosis, *Alliaceae* extract, ionophore, microbiota intestinal

## Abstract

**Simple Summary:**

Coccidiosis is caused by an intracellular parasite that damages the intestinal integrity, negatively affecting the digestion and absorption of nutrients and consequently worsening weight gain, feed efficiency, and pigmentation of birds, even causing mortality. Therefore, it has a negative impact on the economy of the poultry industry. Currently, the disease is mainly treated by using anticoccidials drugs added to the diet. The drug resistance, as well as the residue of drugs in the meat, has prompted the development of natural alternatives to combat coccidiosis. The purpose of this research was to determine whether an *Alliaceae* encapsulated extract added to the broiler chickens diet decreased the number of oocysts excreted in feces and the harm caused to the intestinal mucosa, consequently improving the productive performance of broiler chickens challenged with *Eimeria* spp. Under our experimental conditions, both the inclusion of *Alliaceae* extract, as well as the use of conventional anticoccidials (nicarbazin/narasin/salinomycin), diminished the detrimental effect of *Eimeria* spp. Moreover, the *Alliaceae* extract favored the abundance of acid butyric bacteria (*Ruminococcus* spp. and *Intestinimonas* spp.) in the cecum, related to intestinal health. Based on the current findings, the *Alliaceae* extract could be a natural additive used to lessen the effects of coccidiosis infections.

**Abstract:**

This study analyzed the effects of an *Alliaceae* encapsulated extract (AE-e) on daily gain (ADG), feed intake (ADFI), feed conversion ratio (FCR), oocysts per gram of feces (OPG), intestinal lesion (LS), and microbiota composition in broilers challenged with *Eimeria* spp. A total of 4800 one day Cobb-500 were allotted into 10 treatment groups with 12 replicates of 40 birds in a 2 × 4 + 2 factorial arrangement. The first factor was non-challenged (NC) or challenged (C), the second was four levels of AE-e added in the basal diet, 0 (AE0), 250 (AE250), 500 (AE500), and 750 mg·kg^−1^ (AE750), plus two ionophore controls, non-challenged (NC-Ion) and challenged (C-Ion). No interactions were observed between factors (NC0, NC250, NC500, NC750, C0, C250, C500, and C750), while C-Ion improved FCR at 21 d. The challenge affected negatively ADG and FCR and promoted enteropathogens in cecum. AE750 improved FCR in the finisher and cumulative phases, while C-Ion had fewer total OPG than C0 and C250. Likewise, at 21d, C250, C500, and C-Ion had fewer LS than C0, while at 28 d, C750 showed lower than C-Ion. In the cecum microbiota, C500 had more *Ruminococcus, Firmicutes b,* and *Intestinimonas* than C-Ion. In summary, AE-e showed beneficial results in broilers infected with *Eimeria* spp.

## 1. Introduction

Currently, coccidiosis disease continues to be one of the most serious problems in the commercial broiler poultry industry, resulting in great economic loss all over the world [1]. *Eimeria acervulina, Eimeria maxima,* and *Eimeria tenella* are the main species that cause disease in broiler chickens, impairing both intestinal function and growth performance [2]. To date, coccidiosis prevention has been through the addition of synthetic anticoccidials such as nicarbazin, decoquinate, and zoalene, as well as ionophores [3] such as monensin, lasalocid, salinomycin, narasin, etc., to either the poultry diet or drinking water, or the use of vaccines [4]. However, *Eimeria* spp. has developed drug resistance, causing a loss of effectiveness in the anticoccidials [5]. In addition, there is a global concern about drug resistance and the presence of residual drugs in meat [6] that has prompted the study and development of natural alternatives to prevent or control coccidiosis [7], such as phytochemicals, which are a suitable alternative due to their favorable effects against *Eimeria* spp. [8].

Several works have demonstrated that extracts and the essential oil of *Allium cepa* and *Allium sativum* improved the average daily feed intake (ADFI), average daily gain (ADG), and feed conversion ratio (FCR), as well as intestinal health and carcass quality in broiler chickens [9,10,11]. Likewise, blends of the genus *Allium* with oregano essential oil and eugenol [12], as well as organic acids [8,13], have shown benefits in broiler performance and health. The genus *Allium* has different sulfur compounds that have been studied as additives in animal nutrition [14]. These compounds are conformed by sulfur atoms attached to a cyanate group in cyclic or non-cyclic forms [15]. The most studied are the sulfoxydes s-methyl-L-cysteine (methionine), s-allil-L-cysteine (alliin), s-propenyl-L-cysteine (isoalliin), and s-propyl-L-cysteine (propiin). Propyl propane thiosulfinate (PTS) and propyl propane thiosulfonate (PTSO) are derived from the natural degradation of propiin [16,17]. Unlike other sulfur compounds from the genus *Allium*, PTSO is chemically stable but is insoluble in water. For this reason, it should be provided with a water-soluble carrier to increase its availability and absorption [18,19]. In addition, PTS and PTSO have shown to reduce the viability of *E. acervulina* sporozoites, improving the innate immune response [20], and inhibit the sporulation of *E. tenella* [21], and they are able to promote the growth of beneficial bacteria and decrease intestinal pathobionts [22] in broiler chickens. Recently, Aguinaga-Casanas, et al. [19] conducted a study in vitro that showed that PTSO inhibit the capability of *E. acervulina* sporozoites to penetrate Madin-Darby bovine kidney cells (MDBK cells), concluding that PTSO is a promising alternative to coccidiosis treatment. The *Allium* spp. could be used as an alternative to anticoccidials in broiler production due to its proven benefits. Nevertheless, it is necessary to carry out more studies to understand the mechanism by which they exert their favorable effects, as well as determinate the appropriate inclusion level in the poultry diet and the period of use to enhance broiler performance [23].

Based on the above information, we hypothesize that an *Alliaceae (A. cepa* and *A. sativum)* encapsulated extract (AE-e) used as a feed additive decreases the oocysts per gram in feces (OPG), reduces the intestinal lesion score (LS), and improves productive performance, as well as modulates positively the intestinal microbiota in *Eimeria* challenged birds. Therefore, the aim of this study was to examine the effects of increasing levels of AE-e on ADG, ADFI, FCR, OPG, LS, and intestinal microbiota composition in broiler chickens challenged with a mixture of *Eimeria* spp.

## 2. Materials and Methods

### 2.1. Ethical Standard

The present study was approved on 5 October 2020, by the Animal Welfare and Experimentation Ethics Committee of the National Autonomous University of México SICUAE-DC-2020/3-6 in compliance with the Mexican Official Norm NOM-062-ZOO-1999.

### 2.2. Housing, Animals and Experimental Design

A total of 4800 one-day-old Cobb-500 broiler chickens were housed in 120 4 m^2^ pens separated by wire mesh partitions and new wood shaving litter throughout the period of study of 49 d. The facility temperature was set as 30 °C during the first week using thermostatically controlled propane gas heaters, reducing 2.5 °C each week. After the fourth week, the temperature was controlled through curtains and kept between 18–21 °C. The first 4 d after reception, the chickens had access to 23 h of light; after that, a natural photoperiod was maintained throughout the study. The birds were assigned in a completely randomized experimental design with a 2 × 4 + 2 factorial arrangement. One factor was the challenge level, formed by a non-challenged group (NC) and a challenged group (C) with *Eimeria* spp. The second was 4 levels of AE-e (0, 250, 500, and 750 mg AE-e per each kg of feed). In addition, two ionophore controls, C-Ion and NC-Ion, were used to contrast the challenged and non-challenged AE-e treatments, respectively. The experimental unit was the pen. To prevent cross contamination, the C and NC birds were housed in separate but identical buildings.

### 2.3. Alliaceae Encapsulated Extract Supplementation

We used a concentrated liquid commercial *Alliaceae* extract (Garlicon^TM^; DOMCA S.A.U., Granada, Spain), which has shown positive effects on bird productivity [20,24,25]. It was encapsulated into a dextrin–lecithin matrix and validated by the presence of PTSO, which has a concentration of 12 g·kg^−1^, as determined by gas chromatography–mass spectrometry (Gas chromatograph model 7890A Agilent Technologies Inc., coupled to a simple quadrupole mass detector model 5975C Agilent Technologies Inc., Santa Clara, CA, USA). The retention time of the chromatography peak was indicated for the PTSO according to the databases of the NIST/EPA/NIH Mass Spectra Library, version 1.7 (Gaithersburg, MD, USA). The analysis was carried out in the laboratory of the Center for Research in Applied Sciences and Advanced Technology of the National Polytechnic Institute (IPN, Querétaro, Mexico).

### 2.4. Diets and Experimental Groups

A corn–soybean meal basal diet was formulated to meet the nutritional specifications for the Cobb-500 lineage™ (Table 1). The basal diet was split into five portions to be mixed with the experimental doses of AE-e or ionophore and then were pelleted at 80 °C for 30 s. The broiler chickens were on two phases of feeding, a starter phase (1–21 day of age) and a finisher phase (22–49 day of age). The feed was restricted from 15:00 h to 08:00 h to avoid ascites syndrome, and the water was provided ad libitum.

The birds were assigned to either a NC or C group and feed with the basal diet containing 4 different levels of AE-e as follows: basal diet without AE-e (AE0); basal diet with added 250 mg·kg^−1^ of AE-e (AE250); basal diet with added 500 mg·kg^−1^ of AE-e (AE500); and basal diet with added 750 mg·kg^−1^ of AE-e (AE750). Two positive control treatments, non-challenged ionophore control (NC-Ion) and a challenged ionophore control (C-Ion), both formed by a basal diet with 50 ppm of nicarbazin and 50 ppm of narasin added for the starter phase and 60 ppm of salinomycin for the finisher phase, summarizing a total of 10 treatments (NC0, NC250, NC500, NC750, C0, C250, C500, C750, NC-Ion, C-Ion) with 12 replicates of 40 birds for each one.

### 2.5. Productive Performance

Each pen was monitored for body weight (BW), weight gain (WG), and feed intake (FI) at 0, 21, and 49 d of age. On the same day as the event took place, we recorded the age and weight of dead birds to determine (a) ADG: [(mean final BW of live birds in the pen) − (mean initial BW of all birds in that pen)]/days of testing. (b) ADFI: (total feed consumed in a pen)/(birds alive × days on test in the pen + days dead birds on test in that pen). (c) FCR: (total feed consumption in a pen)/(WG of birds alive + WG of dead birds in the same pen) [26].

### 2.6. Eimeria Challenge 

At 12 d of age, the broiler chickens from the challenged group (C) were inoculated directly into the crop with 0.5 mL of a mixture of sporulated oocysts of *E. acervulina* 1 × 10^5^, *E. maxima* 2 × 10^4^, and *E. tenella* 2 × 10^4^ using sterile plastic syringes, while the birds from the non-challenged group (NC) received a sham 0.5 mL of distilled water.

The *Eimeria* mixture was obtained from a non-governmental laboratory of parasitology, Morelos, México, and it was assessed by counting the oocysts sporulated from the different species of *Eimeria* at the National Autonomous University of México (UNAM) and by PCR Sanger sequencing at the Faculty of Chemistry, Querétaro University, México (UAQ). 

### 2.7. Eimeria Oocysts Count

On d 9, 16, and 23 post-inoculation (p.i.), approximately 10 g of fresh fecal material was collected from each pen and mixed thoroughly in a plastic bag and kept at 4 °C until the total count of oocysts. Five grams of each sample was homogenized in a saturated NaCl solution (400 g·L^−1^) and filtered through a 300-mesh sieve. The filtrate was centrifuged at 800× *g* for 2 min and an aliquot of the supernatant was poured into a Mc Master Chamber and counted at 10× magnification on a compound microscope following the technique described by Long, et al. [27]. The morphological characteristics of the sporulated oocysts were used to identify *E. acervulina*, *E. maxima*, and *E. tenella*; the number of oocysts was expressed as OPG. The total OPG is the sum of OPG for all three species.

### 2.8. Intestinal Lesion Score (LS)

The LS in the duodenum, jejunum, and cecum were evaluated at 9- and 16-day p.i., twenty-four birds from each treatment were randomly selected and humanely killed by cervical dislocation [28,29]. The gastrointestinal tract was removed and opened; the scores for macroscopic lesions for *E. acervulina, E. maxima, and E. tenella* were determined according to the scale of Johnson and Reid [30]. A score of “0” represented no visual lesions, “1” was minimal lesions, “2” was moderate lesions, “3” was severe lesions, and “4” was extremely severe lesions.

### 2.9. Anticoccidial Index (ACI)

The relative ratio weight gain (rBWG), survival rate (SR), total mean lesion score (TMLS), and OPG value are necessary to calculate the anticoccidial index (ACI) and are recognizes as good indicators of the efficacy of the anticoccidial compounds. The ACI was calculated for each group according to the following equation proposed by Merk, et al. [31]:ACI = (rBWG + SR) × 100 − (TMLS × 10 + OPG value)

The variables were calculated as follows:rBWG: BWG rate of the challenged unmedicated control or drug treated group/BWG rate of unchallenged unmedicated control group × 100.BWG rate: (Final BW − initial BW)/initial BW × 100.SR: Number of final birds alive/ number of total initial birds × 100.TMLS: Sum of the LS caused by all the *Eimeria* spp.OPG value: OPG in unchallenged unmedicated control or challenged drug-treated group/OPG in infected/unmedicated control group × 100 [32].

### 2.10. Intestinal Microbiota Samples

At 21 d of age (9 d p.i.), 6 chickens from 5 treatments, NC0, C0, NC500, C500, and C-Ion, were randomly selected and sacrificed by the manual cervical dislocation method [29,33]. The gastrointestinal tract was dissected, the ileum and cecum contents were scraped carefully and collected in cryogenic vials, snap frozen in liquid nitrogen, and stored at −80 °C until the microbiota composition analysis [34,35].

### 2.11. DNA Extraction, 16s rRNA Gene Amplification, and Library Preparation for Sequencing

The bacterial DNA from the ileal and cecal contents was extracted using the ZymoBIOMICS™ DNA Miniprep kit (D4300 Zymo Research, Irvine, CA, USA), according to the manufacturer recommendations, it was quantified by fluorometry using Qubit chemistry (Invitrogen, Waltham, MA, USA), while its integrity was assessed by spectrometry (NanoDrop, Thermo Fisher Scientific, Whaltam, MA, USA). The libraries were made following the two-step polymerase chain reaction (PCR) protocol suggested by Illumina (Illumina Part# 15044223 Rev.B, San Diego, CA, USA) to sequence a single segment comprising the 16S rRNA V3-V4 region [36]. The libraries were quantified by fluorometry, pooled at 4nM with 10% PhiX sequencing control, and sequenced using the Illumina MiSeq platform to obtain 300 paired-end reads following the manufacturer´s instructions (Illumina, San Diego, CA, USA).

### 2.12. Bioinformatic Analysis 

The paired-end raw reads were analyzed with Cutadapt v1.15 to eliminate any traces of 16S-rRNA amplification primers or Illumina adapter sequences and then scanned with Trimmomatic v36 [37] to filter out the lower quality reads. The forward and reverse reads of each pair were then overlapped into single fragments using FLASH v1.2.11 software, employing an expected fragment length of 409 ± 20 bp and an expected read length of 279 bp and were further filtered with DADA2 (included in the QIIME2 v2020.8 suite) [38]; to eliminate reads where 2 or more sequencing errors were expected, groups of reads were produced by experimental errors (noise removal) and chimeric fragments.

To assign a taxonomic classification to the pre-processed sequences, we used the naive Bayes classifier [39], as implemented in the QIIME2 suite [38]. The classifier was trained with the annotations of the SILVA 138 ribosomal reference database [40] using the sequences grouped to a similarity of 99%.

The Shannon and Simpson indexes, as well as the total OTU counts, were obtained for each sample to study the intestinal microbial α-diversity. β-diversity was assessed by measuring the Bray–Curtis dissimilarity of each pair of samples, followed by nonmetric multi-dimensional scaling (NMDS) to observe the clustering of the different sample groups. The α and β-diversity profiles were visualized through box-plots and NMDS scaling plots [41]. To obtain the relative abundance of the OTUs, the number of reads per OTU was normalized by library size. Only the abundance changes in genus with a significance *p* ≤ 0.05 were represented in a heatmap.

### 2.13. Statistical Analyses

The statistical analyses were performed using the JMP statistical software v 17.0.0 (SAS Institute Inc., Cary, NC, USA) and R version 4.0.2. The data normality and variance homogeneity among groups were tested using the Shapiro–Wilk and Levene´s tests, respectively. The variables with non-normal distributions were analyzed by nonparametric statistics. The significance level was set at *p* ≤ 0.05, and a trend was set between *p* > 0.05 and ≤0.10.

For the analysis of productive performance, the initial BW was included as a covariate. The ADFI, ADG, and FCR were analyzed by 2-way ANOVA. The ACI was analyzed by one-way ANOVA. Post hoc Tukey tests were performed. The experimental design considered two controls with ionophores to contrast the treatments. Contrast A was NC-Ion vs. NC0, NC250, NC500, and NC750; contrast B was C-Ion vs. C0, C250, C500, and C750; and contrast C was NC-Ion vs. C-Ion. Additionally, the AE-e factor was analyzed to determine if the effects of different doses of AE-e had a linear trend. Furthermore, AE-e treatments included an analysis of the polynomial orthogonal contrast trend in the variable ACI.

Since there were no detected oocyst or coccidia lesions in nonchallenged birds, the OPG and LS were analyzed only in the challenged birds using the Kruskal–Wallis test and, as post hoc, the Steel–Dwass test, the medians, and quantiles q25 and q75 were reported. Correlations were carried out between ADG, FCR, OPG, and LS using Spearman Rho analysis.

The α-diversity changes among groups were assessed with the Kruskal–Wallis test, while the β-diversity was assessed by the NMDS and PerMANOVA tests to identify significant differences in the clustering position of the groups. To obtain the diversity measures and the corresponding statistical tests, we used the R phyloseq v1.38 package [41]. We used the DESeq2 package to estimate the differential abundance of specific clades between group sample pairs, using the Wald test and adjusting the *p*-values through the Benjamini–Hochberg multiple sampling correction.

## 3. Results

### 3.1. Productive Performance

The following tables provide comprehensive data on the interactions between the *Eimeria* challenge and AE-e supplementation (Table 2) and subsequently present independently the effects of the *Eimeria* challenge (Table 3) and the effects of AE-e supplementation (Table 4) on ADG, ADFI, and FCR. No significant differences (*p* > 0.05) were found in the interaction of factors on ADG, ADFI, and FCR in starter and finisher phases or in the cumulative period of study (Table 2). As well as the orthogonal contrast (contrast A), NC-Ion vs. NC0, NC250, NC500, and NC750 did not show differences in ADFI, ADG, and FCR (*p* > 0.05) in both growing phases. On the other hand, the comparison of contrast B, C-Ion vs. C0, C250, C500 and C750, did not show differences in ADFI and ADG throughout the study period (*p* > 0.05). However, at 21 d, C-Ion showed lower FCR, 1.30 vs. 1.39, respectively, (*p* < 0.01) and a trend (*p* = 0.07) to improve ADG. There was no effect on FCR in the finisher or cumulative period (*p* > 0.05). Finally, in contrast C, NC-Ion vs. C-Ion, there was a trend in the starter phase in which the challenged group (C-Ion) had lower ADG (*p* = 0.06) and higher FCR (*p* = 0.07) than NC-Ion.

At the starter phase, the challenge affected the productive performance. Broiler chickens from the challenged group (C) had a reduction of 12.4% in ADG compared to those from the non-challenged (NC) group (*p* < 0.0001); moreover, the FCR was also deteriorated by 12.8%, 1.40 vs. 1.22 (*p* < 0.0001), respectively. Despite not finding changes in the finisher phase (*p* > 0.05), the negative effect observed in the starter phase continued in the cumulative period, ADG (*p* < 0.001) and FCR (*p* = 0.01) in broiler chickens from C group compared to NC group. However, the ADFI was not affected by the challenge (*p* > 0.05; Table 3).

Table 4 shows that during the starter phase, the inclusion of AE-e did not influence performance (*p* > 0.05). Nevertheless, in the finisher phase, the FCR was better in AE750, 1.97 compared to AE0, 2.20 (*p* = 0.03). Moreover, in the cumulative period, AE750 continued to show better FCR, 1.73 regarding 1.84 from AE0 (*p* = 0.01); however, the AE250 and AE500 inclusion did not show a difference in FCR (*p* > 0.05). In addition, no significant differences in ADG and ADFI were observed between treatments during the study (*p* > 0.05); however, AE500 displayed a trend to improve ADG in the cumulative study (*p* = 0.10). On the other hand, a positive linear trend was observed in ADG (*p* = 0.05) and FCR (*p* = 0.006) in the finisher phase, as well as FCR in the cumulative period (*p* = 0.004).

### 3.2. Oocysts Shedding 

Table 5 describes the effect of feed supplementation with AE-e or ionophores on the OPG of *Eimeria*-challenged broiler chickens (C0, C250, C500, C750, and C-Ion), because of the absence of OPG in non-challenged treatments, NC0, NC250, NC500, NC750, and NC-Ion were omitted from analysis. The differences between treatments were observed at 9 d p.i., corresponding to 21 d of age. The higher OPG values of *E. acervulina* and *E. tenella* were observed in C0, 7.75 and 22.26-fold changes, respectively, regarding C-Ion (*p* < 0.05).

While C250, C500, and C750 had a similar OPG shedding in both species compared to C-Ion and C0 (*p* > 0.05), in addition, *E. maxima* did not show differences between treatments (*p* > 0.05). Moreover, the total OPG values were significantly higher (+5-fold) in C0 and C250 than C-Ion (*p* < 0.01), whereas C500 and C750 had a similar OPG shedding compared to C-Ion, C0, and C250 (*p* > 0.05). The decrease in OPG excretion from 21 to 28 d of age was more evident in the groups C250 (−69.66-fold), C500 (−69.85-fold), and C750 (−30.73-fold) than C-Ion (−1.94-fold) and C0 (−27.11-fold). At 28 and 35 day of age, *E. acervulina, E. maxima*, and *E. tenella* OPG differences between treatments were not detected (*p* > 0.05).

### 3.3. Intestinal Lesion Score 

Due to the absence of injuries in the NC groups, NC0, NC250, NC500, NC750, and NC-Ion, we only show the analysis of the challenged groups, C0, C250, C500, C750, and C-Ion. The LS of duodenum, jejunum, and cecum, as well as TMLS, are described in Table 6. At 21 days of age (9 d p.i.), the LS observed in the duodenum reveled that the birds given the C0 diet showed a more severe LS of 1.5, which was significantly (*p* < 0.0001) higher than the 0.5, 1, and 0 scores from birds given the C250, C500, and C-Ion diets, respectively. Remarkably, the LS observed in broilers from the C250 diet were similar to those observed in the broilers on the C-Ion diet. In addition, no significant differences (*p* > 0.05) were detected among the C250, C500, and C750 diets. However, in the jejunum, significant differences were identified (*p* = 0.03) between the C0 and C750 vs. the C250 and C-Ion diets; despite the LS medians being equal in the groups, the range q25–q75 was higher in C0 and C750, (0-1), than in C250 and C-Ion, (0-0). In addition, in this intestinal section, the C500 diet had similar effects to the others on LS (*p* > 0.05). In the cecum, the LS in the broilers given the C-Ion diet were lower than those observed in birds fed the C0, C250, and C750 diets (*p* < 0.001) but not compared to C500 (*p* > 0.05). Furthermore, on day 21, the total mean lesion score (TMLS) was significantly higher (*p* < 0.001) in birds from diet C0 compared to C250, C500, and C-Ion. Moreover, the chicken on the C-Ion diet had lower TMLS compared to other treatments at 21 d of age (*p* < 0.0001). On the other hand, on day 28, the LS caused by *E. acervuline, E. maxima,* and *E. tenella* were not different between all treatments (*p* > 0.05). However, C-Ion showed higher TMLS compared to the C750 diet (*p* = 0.02), while the C0, C250, and C500 diets did not show significant differences (*p* > 0.05).

### 3.4. Anticoccidial Index (ACI) 

Table 7 describes the ACI of the different treatments, NC0 had superior ACI compared to C0, C250, C500, and C750 (*p* < 0.0001). C-Ion performed better than C0 and C250 (*p* < 0.0001). The groups C500 and C750 had similar results compared to C-Ion, C0, and C250 (*p* > 0.05). There were linear (*p* < 0.04) and quadratic (*p* < 0.03) positive trends in ACI when comparing the responses to C0, C250, C500 and C750.

Spearman correlations (rho) were calculated between OPG, TMLS, ADG, and FCR; the coefficient showed that the presence of OPG at 21 d of age had a positive correlation of 0.80 and 0.71 with TMLS at 21 and 28 d of age, respectively (*p* < 0.0001). In addition, this OPG had also a positive correlation of 0.73 (*p* < 0.0001) with the FCR at 21 d. On the other hand, there was a negative correlation (−0.54) with ADG at 21 d of age and −0.29 at 49 d of age (*p* < 0.01). At 28 d of age, OPG and TMLS showed a positive correlation of 0.42 (*p* < 0.001). The TMLS at 21 and 28 d were positively correlated (0.67). TMLS at 21d showed a positive correlation of 0.72 with the FCR in the starter phase (*p* < 0.001), as well as a positive correlation of 0.22 with the cumulative period (*p* < 0.05), while showing a negative correlation (−0.51) with the ADG at the same age (*p* < 0.001) and of −0.18 with ADG at 49 d (*p* < 0.05). Therefore, TMLS at 28 d was negatively correlated (−0.18) with ADG at 49 d of age (*p* < 0.05).

### 3.5. Analysis of Bacterial Composition, 16s rRNA

To evaluate the effects of 5 treatments, NC0, NC500, C0, C500, and C-Ion, on the intestinal microbiota composition in broiler chickens, we studied the ileum and cecum microbial community using 16S rRNA sequencing. We obtained a total of 2,683,053 high-quality sequences of the V3–V4 region of the 16S rRNA gene; a total of 10,710 operational taxonomic units (OTUs) at the 99% sequence similarity level were identified in all samples. 

### 3.6. Alpha- and Beta-Diversity

We detected no differences in the α-diversity indexes (Shannon and Simpson) or in the OTU count of different treatment pairs in the ileum and cecum (*p* > 0.05). In addition, the β-diversity assessed by NMDS from Bray–Curtis distance matrices and a permutational analysis of variance in the ileum section showed no differences in the microbiome between the treatments (*p* > 0.05). On the other hand, in the cecum, there were significant differences in the clustering position in the groups at the taxonomic levels of family, Figure 1a (*p* < 0.01), and genus, Figure 1b (*p* = 0.01). The results showed that NC0 and NC500 were different from C0 and C500, while C-Ion showed no differences against any treatment, except in the taxonomic category family, which was different from NC0, Figure 1.

### 3.7. Relative Intestinal Microbiota Abundance

Figure 2 depicts the analysis of the relative abundance of different bacterial clades in the cecum at the phylum and genus level for each group of samples. In the cecum microbiota, *Firmicutes* was the dominant phylum (Figure 2a), with a relative abundance ranging from 57.8% (C0 treatment) to 79.4% (NC0 treatment). At the genus level (Figure 2b), *Bacteroides* was the taxon with the highest relative abundance (9.7–22.1%), highlighting their presence in the groups challenged, C0, C500, and C-Ion.

To further explore the effect of the treatments on the cecum bacterial communities, we searched for bacterial genus abundance changes among pairs of treatments, C0 vs. NC0; C500 vs. NC500; C-Ion vs. C0; NC500 vs. NC0; C500 vs. C0; and C500 vs. C-Ion (Figure 3). The results are presented in a heat map from a hierarchical clustering analysis based only on significant changes (*p* < 0.05). The results from this study showed that the pair comparisons C0 vs. NC0 and C500 vs. NC500 had more bacterial change-in-abundance differences than the other comparisons studied: C-Ion vs. C0; NC500 vs. NC0; C500 vs. C0; and C500 vs. C-Ion. There was a significantly higher abundance (*p* < 0.05) of genus *Ruminococcus 2* in birds from C500 compared to NC500, C0, and C-Ion (+23.36 log2-fc, +8.0 log2-fc, and +23.08 log2-fc, respectively). Moreover, the abundance change in *Intestinimonas* in the C500 group was +1.78 log2-fc higher than the C-Ion group. However, *Escherichia-Shigella*, another predominant bacterium in broiler chicken intestines, was also found to be abundant (+4.2 log2fc) in C0 with respect to the NC0 (*p* < 0.05). Other bacteria, such as *Tyzzerella*, *Eggerthella*, *Clostridium innocuum* g., *Ruminococcaceae* UCG-009, u. *Clostridia* b., and *Ruminococcus* 1., show a higher abundance under the *Eimeria* challenge conditions. On the other hand, NC0 has higher numbers (*p* < 0.05) of *Bacillus, Hydrogenoanaerobacterium,* and *Ruminococus 1* than its challenged counterparts.

## 4. Discussion

In our study, the *Eimeria* challenge caused a negative impact on growth performance parameters, being more evident at 9 d p.i. (21 days of age). The decrease in the ADG in infected birds, as well as the increases in the FCR, has been reported in other works [8,26,42,43]. In our study, the inclusion of AE750 improved the FCR of broilers in the finisher phase, and in the cumulative study regarding AE0, these results are in agreement with Peinado, et al. [44], who also include two levels of PTSO in the broiler chicken diet and did not find any effect on ADFI, whereas ADG improved using 45 mg per PTSO kg^−1^ of the diet. Moreover, the FCR was enhanced with both doses tested, 45 and 90 mg per PTSO kg^−1^ of diet. Similar results were obtained by Kim, et al. [20], who reported that broilers challenged with *E. acervuline* and fed with 10 ppm of PTSO (67%) and PTS (33%) showed a better growth than the birds that were not supplemented (*p* < 0.05). Furthermore, Kairalla, et al. [11] reported that some feed additives, particularly garlic (*A. sativum*), have shown to improve FCR, as was also previously demonstrated by Aarti and Khusro [45]. This result is important since *Eimeria* infection destroys epithelial cells and affects intestinal villi, causing poor nutrient digestion and severe damage to the host intestinal mucosa, resulting in clinical or subclinical symptoms [46]. In this regard, in a previous study, we demonstrated that organosulfur compounds from garlic, particularly PTSO in 250 g·t^−1^ of feed, improved the amino acid and energy digestibility in broiler chickens fed with a soybean meal–yellow corn diet [47].

ADG or FCR are not the only good indicators for measuring the effectiveness of anticoccidial drug, but LS and OPG are also considered complementary indicators [48] to the performance. In our study, the higher OPG excretion detected was 9 d p.i., then OPG gradually decreased over time; at 16 d p.i., it is barely noticeable and almost disappears at 23 d p.i. This trend after infection was reported by You [49]. In addition to the treatment effect, the reduction in OPG may be due in part to self-limitation of parasitosis and the immune response developed by the host [50]. We found that OPG shedding was decreased in the challenged birds supplemented with anticoccidial treatment (C-Ion), while 500 and 750 mg·kg^−1^ of AE-e tended to reduce OPG shedding. A similar effect was observed on the OPG of broiler chickens challenged with *Eimeria* and supplemented with garlic extracts (*A. sativum*) [51,52], their active derivative compounds [20], or a premix of garlic and oregano essential oils [53]. The ACI results were better for C-Ion than for C0 and C250. A positive linear and quadratic trend in ACI suggests that generally, birds challenged with *Eimeria* spp. would perform better when 500 or 750 mg·kg^−1^ of AE-e is included in their diet.

The correlation analysis in our study indicates that the reduction in the LS was supported by the decreasing OPG. Similar results were found by Elkhtam, et al. [51], where OPG, the clinical symptoms of the disease, and the LS decreased with the addition of garlic extract (*A. sativum*) to the diet of the broiler chickens challenged with *Eimeria* spp. The LS could explain the negative effect on ADG and FCR at 21 and 28 d of age, which was previously noticed by Reid and Johnson [54], who contrasted the LS of birds infected with *E. acervulina* with their weight reached, finding that at higher LS, a lower ADG was obtained, mostly by one week after the challenge; however, LS are not always correlated in coccidia infections, as reported by Ringenier, et al. [55], who did not find a relationship between LS and FCR in broilers at 28 d of age, concluding that broiler chickens are able to cope with a certain level of gut damage before it influences the overall performance. Other researchers such as Conway, et al. [56] concluded that the correlation between OPG, LS, ADG, and FCR depends on the type of *Eimeria* and the use or not of anticoccidials.

On the other hand, it has been reported that diet composition [57,58] and phytochemicals modulate intestinal microbiota [44,53]. It is well known that a healthy and functional intestinal microbiome is related to a positive productive performance of the chickens [59,60]. It was reported that when growing broiler chickens that feed on diets supplemented with *Allium* derivatives, PTSO or PTS are able to influence intestinal microbiota composition [22,61], decreasing enteropathogens and increasing the nutrient absorption in the intestine [22,62,63,64]. In the current study, neither AE-e nor anticoccidials modified the α- or β-diversity in the ileum or the α-diversity in the cecum. Similar findings were noted by Abdelli, et al. [13], who found that dietary supplementation with natural compounds such as organic acids and essential oils does not always result in changes in diversity in microbial populations within the gastrointestinal tract. Nevertheless, in the cecum, we found changes in β-diversity between clades at the taxonomic levels of family and genus, where the challenged groups showed a different spatial distribution to those not challenged, concluding that the infection with *Eimeria* spp. influences β-diversity, as reported by other researchers [33,65,66]. The analysis of the microbiota clearly showed that the groups under challenge of *Eimeria* spp., regardless of the presence of anticoccidial drugs or AE-e in the diet, had a higher number of enteropathogens belonging to the *Enterobacteriaceae* family such as *Proteus*, *Escherichia*, and *Shigella*, as well as other opportunistic pathogens, including the genera *Tyzerella*, *Eggertella,* and *Biophila*.

In our study, we found that in the cecum, at the genus level, C500 had a higher abundance of *Ruminococcus*, *Firmicutes* b., and *Intestinimonas* than C-Ion, considering that *Ruminococcus* bacteria synthesize digestive enzymes such as cellulases, xylanases, and cellobioses [67,68,69], which contributes to the hydrolysis and fermentation of non-structural carbohydrates, producing butyric acid. It has also been reported that some species of *Ruminococcus* produce bacteriocins that contribute to controlling undesirable bacterial populations, enhancing the growth of *Lactobacillus* and promoting intestinal health. All this could explain the improvement in nutrient digestibility and the productive behavior of birds when *Ruminococcus* is present in abundance. On the other hand, species of the *Intestinimonas* genus are producers of short-chain fatty acids increasing butyric fatty acid from simple sugars and amino acids such as lysine [70,71]. It connects two important metabolic characteristics, butyric acid production and amino acid fermentation in the intestinal tract. Thus, in this study, the beneficial effects of AE-e on the modulation of the intestinal microbiota were consistent with the results of Vezza, et al. [57], who demonstrated in a murine model of metabolic syndrome that PTSO supplementation at doses of 0.1, 0.5, and 1 mg·kg^−1^·day^−1^ counteracts intestinal dysbiosis.

## 5. Conclusions

In summary, the dose of 750 mg·kg^−1^ of an *Alliaceae* encapsulated extract added to the diet of broiler chickens improved the feed conversion ratio in the finisher phase compared to broiler chickens fed a diet without additives. Furthermore, during the finisher phase and cumulative study, its addition to the diet resulted in a positive linear trend in average daily gain and feed conversion ratio.

The anticoccidial index showed a quadratic trend in which the dose of 500 mg·kg^−1^ of the *Alliaceae* encapsulated extract displayed the best response. Moreover, it promoted the abundance of some butyrate-producing bacteria such as *Intestinimonas* and *Ruminococcus* in the cecum.

However, the best anticoccidial index, feed conversion ratio, oocyst shedding, and intestinal lesion score was observed in the group fed with the anticoccidial ionophore program.

Further research is required to explain the mode of action, as well as determine the optimal dose of the *Alliaceae* encapsulated extract in the diet of broiler chickens to lessen or control the detrimental effects of coccidiosis under industrial conditions. It is necessary to verify whether using an *Alliaceae* encapsulated extract in dual programs combined with ionophores or vaccines is feasible.

## Figures and Tables

**Figure 1 animals-13-03884-f001:**
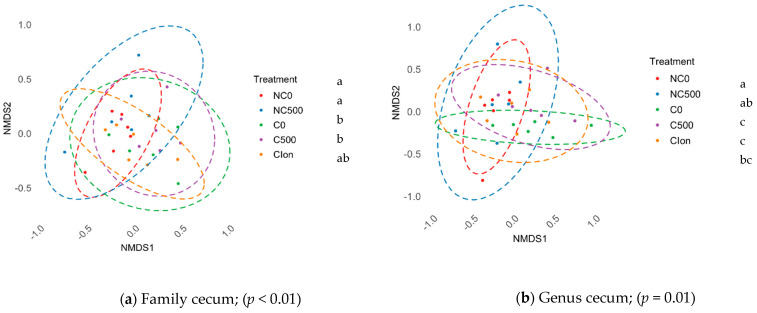
β-diversity comparison between treatments ^1^. NMDS of Bray-Curtis distances in (**a**) Family cecum and (**b**) Genus cecum. ^abc^ The treatments not sharing superscript are different (*p* < 0.05). NC0, Non-challenged + basal diet; NC500, Non-challenged + basal diet + 500 mg·kg^−1^ AE-e; C0, Challenged + basal diet; C500, Challenged + basal diet + 500 mg·kg^−1^ AE-e; C-Ion, Challenged + basal diet + 50 ppm nicarbazin–50 ppm narasin. Challenged with *E. acervulina* 1 × 10^5^, *E. maxima* 2 × 10^4^, and *E. tenella* 2 × 10^4^.

**Figure 2 animals-13-03884-f002:**
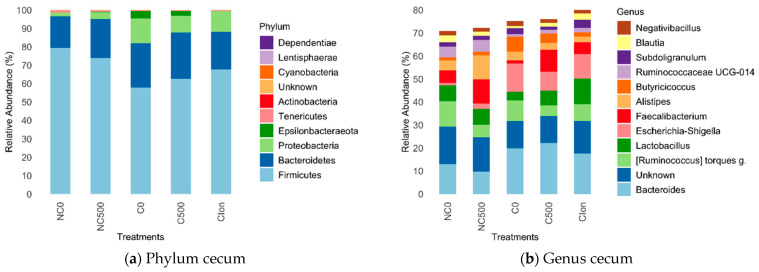
Relative abundance in the intestinal microbiota in broiler chickens at 21 d in (**a**) Phylum cecum, (**b**) Genus cecum. Treatments: NC0, Non-challenged + basal diet; NC500, Non-challenged + basal diet + 500 mg·kg^−1^ AE-e; C0, Challenged + basal diet; C500, Challenged + basal diet + 500 mg·kg^−1^ AE-e; C-Ion, Challenged + basal diet+50 ppm nicarbazin–50 ppm narasin. Challenged with *E. acervulina* 1 × 10^5^, *E. maxima* 2 × 10^4^, and *E. tenella* 2 × 10^4^. The relative abundances were obtained after normalizing the reads per clade counts by sequencing the library size and obtaining the average for each clade across treatments. The 12 most abundant clades are shown.

**Figure 3 animals-13-03884-f003:**
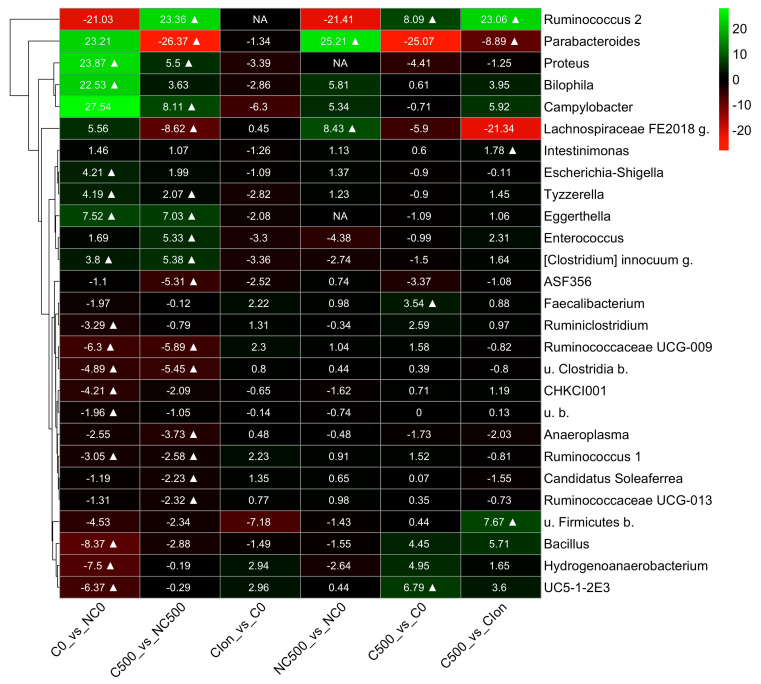
Heat map of the bacterial genus from the cecum of broiler chickens at 21d (9d p.i.). The color block in the heatmap indicates the normalized abundance log2 fold change (log2-fc) of each pair of groups at the bacterial genus: C0 vs. NC0; C500 vs. NC500; C-Ion vs. C0; NC500 vs. NC0; C500 vs. C0; and C500 vs. C-Ion. The positive and negative changes are indicated by the intensity of the green and red color, respectively. Significant changes (with an adjusted *p* ≤ 0.05) are marked with a symbol ∆. The non-available (NA) label indicates that the corresponding comparison was not performed as there were no counts for most of the samples of the group pair. A hierarchical clustering tree based on the log2-fc signals of the clades is shown to the left of the heatmap. The abbreviations u. b. and g. in the genus names stand for “unknown”, “bacterium”, and “genus”, respectively.

**Table 1 animals-13-03884-t001:** Ingredients (kg·t^−1^ of feed) and calculated chemical composition (% as fed) and metabolizable energy (EM, kcal·kg^−1^) in the basal diets.

Ingredient	Starter (0–21 d)	Finisher (22–49 d)
Corn	513.9	552.5
Soybean meal	406.0	360.0
Soybean oil	40.5	51.0
Limestone	14.7	13.6
Calcium orthophosphate	9.1	7.0
Sodium carbonate	4.9	4.7
Methionine DL	3.6	3.2
Xanthophylls.		2.4
Salt	2.0	1.6
L-Lysine HCL	2.2	1.5
Threonine	1.1	0.8
Vitamin-and mineral trace Premix ^1^	0.9	0.8
Betaine anhydrous	0.6	0.4
L-valine	0.2	
Biocholine	0.2	0.17
Tryptophan		0.16
Phytase 5000	0.1	0.12
**Calculated nutrient levels (%)**		
Humidity	11.61	11.66
EM (kcal/kg)	3156	3245
Crude protein CP	23.90	21.60
Ether extract	6.21	7.34
Ashes	5.83	5.27
Crude fiber	2.34	2.47
Total phosphorous	0.59	0.53
Total calcium	1.00	0.90
Sodium	0.23	0.21
Lysine	1.50	1.29
Xanthophylls	0.0008	0.0080

^1^ Content per kilogram: vitamin A (retinol acetate), 12,000 international units (IU); vitamin D3, 5000 IU; vitamin E (DL-α-tocopherol acetate), 50 IU; vitamin K, 3 mg; thiamine, 3 mg; riboflavin, 9 mg; pantothenic acid, 15 mg; pyridoxine, 4 mg; biotin, 0.2 mg; folic acid, 2 mg; vitamin B12, 0.02 mg; manganese, 100 mg; zinc 100 mg; iron, 40 mg; copper, 15 mg; iodine, 1 mg; selenium, and 0.35 mg.

**Table 2 animals-13-03884-t002:** Initial body weight (BW), average daily gain (ADG), average daily feed intake (ADFI), and feed conversion ratio (FCR) of broilers chickens fed a corn–soybean diet supplemented with different doses of *Alliaceae* encapsulated extract (AE-e) ^1^ or anticoccidial drugs (Ion) ^2^ under challenge with *Eimeria* spp. ^3^.

													Contrasts ^4^		
	Non-Challenged (NC)				Challenged (C)								A	B	C
	NC0	NC250	NC500	NC750	C0	C250	C500	C750	SEM	*p* val.	NC-Ion	C-Ion	*p* val.	*p* val.	*p* val.
BW 0 d	42.3	41.9	41.7	41.9	42.1	42.3	42.2	42.0	0.21	0.30	42.0	42.2	0.71	0.69	0.53
Starter phase (1–21 d)															
ADG (g)	48.2	47.0	47.4	47.5	41.3	41.7	43.5	39.9	1.09	0.33	46.5	43.7	0.41	0.07	0.06
ADFI (g)	58.4	58.1	58.1	58.2	58.7	59.9	58.7	57.2	0.81	0.36	58.1	57.3	0.90	0.15	0.47
FCR	1.21	1.24	1.22	1.21	1.41	1.43	1.34	1.41	0.03	0.47	1.23	1.30	0.75	0.01	0.07
Finisher phase (22–49 d)															
ADG (g)	79.5	81.5	84.0	82.4	78.2	80.6	82.2	82.7	1.99	0.96	84.1	82.6	0.29	0.42	0.58
ADFI (g)	156.6	159.1	158.6	149.6	157.0	156.2	151.9	151.7	3.95	0.70	150.9	153.1	0.20	0.87	0.67
FCR	2.16	2.06	2.12	2.01	2.23	2.08	1.97	1.93	0.08	0.47	1.99	2.00	0.21	0.58	0.92
Cumulative study (1–49 d)															
ADG (g)	66.1	66.7	68.3	67.4	62.4	63.9	65.6	64.4	1.10	0.97	68.0	66.0	0.46	0.11	0.17
ADFI (g)	114.5	115.8	115.5	110.5	114.8	115.0	112.0	111.2	2.32	0.80	111.1	112.1	0.20	0.70	0.76
FCR	1.77	1.75	1.75	1.70	1.91	1.84	1.74	1.76	0.03	0.22	1.69	1.74	0.17	0.19	0.29

^1^ NC0, Non-challenged + basal diet; NC250, Non-challenged + basal diet + 250 mg·kg^−1^ AE-e; NC500, Non-challenged + basal diet + 500 mg·kg^−1^ AE-e; NC750, Non-challenged + basal diet + 750 mg·kg^−1^ AE-e; C0, Challenged + basal diet; C250, Challenged + basal diet + 250 mg·kg^−1^ AE-e; C500, Challenged + basal diet + 500 mg·kg^−1^ AE-e; C750, Challenged + basal diet + 750 mg·kg^−1^ AE-e; ^2^ NC-Ion, Non-challenged + basal diet + 50 ppm nicarbazin–50 ppm narasin 1–21 d/ salinomycin 60 ppm 22–49 d. C-Ion, Challenged + basal diet + 50 ppm nicarbazin–50 ppm narasin 1–21 d/ salinomycin 60 ppm 22–49 d. ^3^ Challenge with *E. acervulina* 1 × 10^5^, *E. maxima* 2 × 10^4^, and *E. tenella* 2 × 10^4^, ^4^ Contrast A: NC-Ion vs. NC0, NC250, NC500, and NC750; Contrast B: C-Ion vs. C0, C250, C500, and C750; Contrast C: NC-Ion vs. C-Ion.

**Table 3 animals-13-03884-t003:** Effects of the challenge with *Eimeria* spp. ^1^ on average daily gain (ADG), average daily feed intake (ADFI), and feed conversion ratio (FCR) of broiler chickens fed a corn–soybean diet.

	Challenge Factor			
	NC	C	SEM	*p* Value
Body weight (0 d)	42.00	42.10	0.11	0.23
Starter phase				
ADG (g)	47.50 ^a^	41.60 ^b^	0.54	<0.001
ADFI (g)	58.20	58.60	0.40	0.44
FCR	1.22 ^a^	1.40 ^b^	0.01	<0.001
Finisher phase				
ADG (g)	81.80	80.90	0.99	0.53
ADFI (g)	156.00	154.20	1.97	0.52
FCR	2.09	2.05	0.04	0.51
Cumulative study				
ADG (g)	67.10 ^a^	64.10 ^b^	0.55	<0.001
ADFI (g)	114.10	113.20	1.17	0.61
FCR	1.74 ^a^	1.81 ^b^	0.02	0.01

^ab^ Means within rows not sharing a common superscript differ at the *P* value reported. ^1^ NC Non-challenged; C, challenged with *E. acervulina* 1 × 10^5^, *E. maxima* 2 × 10^4^, and *E. tenella* 2 × 10^4^.

**Table 4 animals-13-03884-t004:** Initial body weight (BW), average daily gain (ADG), average daily feed intake (ADFI), and feed conversion ratio (FCR) of broiler chickens fed a corn–soybean diet supplemented with different doses of *Alliaceae* encapsulated extract (AE-e) ^1^.

	AE-e Factor						Linear Trend ^2^	
	AE0	AE250	AE500	AE750	SEM	*p* Value	*p* Value	R^2^
BW 0 d	42.20	42.10	42.00	42.00	0.15	0.65		
Starter phase								
ADG (g)	44.80	44.40	45.54	43.70	0.77	0.41	0.61	0.002
ADFI (g)	58.60	59.00	58.40	57.70	0.57	0.45	0.38	0.008
FCR	1.31	1.34	1.28	1.31	0.02	0.23	0.82	0.005
Finisher phase								
ADG (g)	78.90	81.00	83.10	82.60	1.40	0.15	0.05	0.04
ADFI (g)	156.70	157.70	155.30	150.70	2.78	0.31	0.16	0.02
FCR	2.20 ^a^	2.07 ^ab^	2.05 ^ab^	1.97 ^b^	0.05	0.03	0.006	0.08
Cumulative study								
ADG (g)	64.20	65.30	67.00	65.90	0.78	0.10	0.14	0.02
ADFI (g)	114.70	115.40	113.80	110.80	1.64	0.23	0.13	0.02
FCR	1.84 ^a^	1.79 ^ab^	1.75 ^ab^	1.73 ^b^	0.03	0.01	0.004	0.08

^ab^ Means within rows not sharing a common superscript differ at *p* value reported. ^1^ AE0, basal diet; AE250, basal diet + 250 mg·kg^−1^ AE-e; AE500, basal diet + 500 mg·kg^−1^ AE-e; AE750, basal diet + 750 mg·kg^−1^ AE-e; ^2^ Linear trend AE0, AE250, AE500, and AE750.

**Table 5 animals-13-03884-t005:** Medians and ranges (Q25–Q75) of oocysts per gram of feces (OPG) in broiler chickens supplemented with different doses of *Alliaceae* encapsulated extract (AE-e) ^1^ or anticoccidial drugs (Ion) ^2^, under challenge with *Eimeria* spp. ^3^.

		Challenge Treatments						
OPG	Age (d)	C0	C250	C500	C750	C-Ion	H ^4^ (Ji^2^)	*p* Value
Total OPG	21	95,575 ^a^ (35,712–186,537)	109,725 ^a^ (28,150–125,350)	48,900 ^ab^ (35,887–66,337)	35,350 ^ab^ (24,462–107,225)	17,975 ^b^ (4862–56,237)	14.22	<0.01
	28	3525 (112–6787)	1575 387–17,612)	700 (0–12,262)	1150 (0–3000)	9225 (812–17,275)	3.76	0.44
	35	0 (0–0)	0 (0–187)	0 (0–37)	0 (0–0)	0 (0–225)	5.50	0.24
*E* *acervulina*	21	78,700 ^a^ (30,537–142,175)	62,850 ^ab^ (12,925–86,512)	32,900 ^ab^ (24,300–46,575)	31,700 ^ab^ (18,550–83,950)	10,150 ^b^ (1562–49,687)	12.16	0.02
	28	2625 (50–5350)	1025 (50–15,350)	375 (0–9000)	400 (0–1200)	7950 (200–16,400)	5.20	0.26
	35	0 (0–0)	0 (0–0)	0 (0–0)	0 (0–0)	0 (0–100)	10.36	0.35
*E. maxima*	21	3750 (387–16,550)	850 (25–11,312)	3925 (62–14,450)	1550 (162–6037)	3500 (1437–7462)	2.77	0.60
	28	0 (0–137)	25 (0–187)	0 (0–175)	0 (0–250)	100 (0–725)	3.11	0.53
	35	0 (0–0)	0 (0–187)	0 (0–0)	0 (0–0)	0 (0–37)	2.79	0.60
*E. tenella*	21	21,150 ^a^ (3550–35,862)	15,450 ^ab^ (0–54,025)	3875 ^ab^ (287–17,425)	2575 ^ab^ (637–10,700)	950 ^b^ (62–5675)	10.33	0.04
	28	700 (25–1462)	325 (212–1225)	225 (0–8712)	100 (0–2400)	425 (25–1950)	0.51	0.97
	35	0 (0–0)	0 (0–0)	0 (0–0)	0 (0–0)	0 (0–0)	2.80	0.59

^ab^ Medians within rows not sharing a common superscript differ at *P value* reported. ^1^ C0, Challenged + basal diet; C250, Challenged + basal diet + 250 mg·kg^−1^ AE-e; C500, Challenged + basal diet + 500 mg·kg^−1^ AE-e; C750, Challenged + basal diet + 750 mg·kg^−1^ AE-e; ^2^ C-Ion, Challenged + basal diet + 50 ppm nicarbazin–50 ppm narasin 1–21 d/ salinomycin 60 ppm 22–49 d. ^3^ Challenged with *E. acervulina* 1 × 10^5^, *E. maxima* 2 × 10^4.^ and *E. tenella* 2 × 10^4^. ^4^ H, test statistic of Kruskal–Wallis.

**Table 6 animals-13-03884-t006:** Medians and ranges (Q25–Q75) in intestinal lesion scores (LS) in broiler chickens supplemented with different doses of *Alliaceae* encapsulated extract (AE-e) ^1^ or anticoccidial drugs (Ion) ^2^, under challenge with *Eimeria* spp.^3^.

		Challenge Treatments						
LS	Age (d)	C0	C250	C500	C750	C-Ion	H ^5^ (Ji^2^)	*p* Value
Duodenum	21	1.5 ^c^ (1–2)	0.5 ^ab^ (0–1)	1 ^b^ (0–1)	1 ^bc^ (1–1)	0 ^a^ (0–0)	49.7	<0.0001
	28	0 (0–1)	0 (0–0)	0 (0–0)	0 (0–0)	0 (0–0.7)	9.2	0.06
Jejunum	21	0 ^b^ (0–1)	0 ^a^ (0–0)	0 ^ab^ (0–0.7)	0 ^b^ (0–1)	0 ^a^ (0–0)	10.7	0.03
	28	0 (0–0)	0 (0–0)	0 (0–0)	0 (0–0)	0 (0–0)	9.7	0.09
Cecum	21	1 ^c^ (1–2)	1 ^bc^ (0–3)	1 ^abc^ (0–1)	1 ^bc^ (0–2)	0 ^a^ (0–0)	31.4	<0.0001
	28	0.5 (0–2)	0.5 (0–2)	1 (0–1)	0 (0–0.7)	1.5 (0–2)	9.2	0.06
TMLS ^4^	21	3 ^c^ (2–4.7)	2 ^b^ (1–3)	2 ^b^ (1–3)	2 ^bc^ (1.2–3.7)	0 ^a^ (0–0.7)	54.0	<0.0001
	28	1 ^ab^ (1–2.7)	1 ^ab^ (0–2)	1 ^ab^ (0–2)	0 ^a^ (0–1.7)	2 ^b^ (1–2)	12.0	0.02

^abc^ Medians within rows not sharing a common superscript differ at *p* value reported. ^1^ C0, Challenged + basal diet; C250, Challenged + basal diet + 250 mg·kg^−1^ AE-e; C500, Challenged + basal diet + 500 mg·kg^−1^ AE-e; C750, Challenged + basal diet + 750 mg·kg^−1^ AE-e; ^2^ C-Ion, Challenged + basal diet + 50 ppm nicarbazin–50 ppm narasin 1–21 d/salinomycin 60 ppm 22–49 d. ^3^ Challenged with *E. acervulina* 1 × 10^5^, *E. maxima* 2 × 10^4.^ and *E. tenella* 2 × 10^4^; ^4^ TMLS Total mean lesion score. ^5^ H, test statistic of Kruskal–Wallis.

**Table 7 animals-13-03884-t007:** Calculated anticoccidial index (ACI) in 21 d broiler chickens supplemented with different doses of *Alliaceae* encapsulated extract (AE-e) ^1^ or anticoccidial drugs (Ion) ^2^, under challenge with *Eimeria* spp. ^3^, as well as orthogonal polynomial contrast comparison of AE-e ^4^.

		Treatments							Linear Trend		Quadratic Trend	
Means ^5^	NC0	C0	C250	C500	C750	C-Ion	SEM	*p* Value	*p* Value	R^2^	*p* Value	R^2^
rBWG %	100	85.8	86.1	90.0	82.7	91.4						
SR %	95.0	91.4	92.1	91.0	93.8	94.8						
TMLS × 10	0	34.2	21.7	19.2	25.8	2.5						
OPG value %	0	100	79.1	54.3	59.3	28.0						
ACI	195 ^a^	43.1 ^c^	77.5 ^c^	107.6 ^bc^	91.3 ^bc^	155.7 ^ab^	21.99	<0.0001	0.03	0.09	0.04	0.13

^abc^ Means within rows not sharing a common superscript differ at *p* value reported. ^1^ NC0, Non-challenged + basal diet; C0, Challenged + basal diet; C250, Challenged + basal diet + 250 mg·kg^−1^ AE-e; C500, Challenged + basal diet + 500 mg·kg^−1^ AE-e; C750, Challenged + basal diet + 750 mg·kg^−1^ AE-e; ^2^ C-Ion, Challenged + basal diet + 50 ppm nicarbazin–50 ppm narasin 1–21 d. ^3^ Challenged with *E. acervulina* 1 × 10^5^, *E. maxima* 2 × 10^4^, and *E. tenella* 2 × 10^4^; ^4^ Polynomic trends, treatments C0, C250, C500, and C750. ^5^ rBWG, relative ratio growth gain; SR, survival rate; TMLS, total mean lesion score; OPG value, oocyst per gram of feces value.

## Data Availability

The data presented in this study are available on request from the corresponding authors. The data are not publicly available due to privacy.

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
