# Peer review of "Effect of an *Alliaceae* Encapsulated Extract on Growth Performance, Gut Health, and Intestinal Microbiota in Broiler Chickens Challenged with *Eimeria* spp."

_animals, 2023, doi:10.3390/ani13243884_

Round 1
Reviewer 1 Report
Comments and Suggestions for Authors
Author Response
Dear reviewer, we want to thank you for the time and dedication you have given to the review of our manuscript, undoubtedly your comments have been very useful to make a better proposal both in form and in substance. We prepared two files, the first of which is where we answer your questions and the second is the corrected version of the paper with the response to the comments to the three reviewers, unfortunately I do not know how to add both files, I am sending just the version of the manuscript which answer all the questions of the 3 reviewers, and in the next paragraph some answers of your main suggestions, we hope that meet your expectation.
The simple summary presents a brief introduction to the topic and objectives of the work, however, according to the journal's standards, it should present a clear statement of the problem addressed, the goals and objectives, pertinent results, conclusions of the study and how they will be valuable to society. I suggest this simple summary be rewritten.
We write the summary again, now the goal and objectives are clear, we add the main results and conclusions, and we highlight the value for society by proposing a natural and harmless product in the diet of broilers to combat coccidiosis.
Use mg.Kg-1, Kg-1, g.Kg-1, kcal.kg-1., g.L-1
All of them are now edited in the correct format
It is necessary to update the references used in the introduction.
Done, now we have 21 of 29 references between 2018 and 2023. The previous manuscript had 12 of 41 in the same range.
As the scientific names have already been presented in the text, the genus can be abbreviated: A. sativum and A. cepa
Done, Now are in Lines 108, 551, 575, and 626
The methods, in general, are described in sufficient detail to allow others to replicate and develop the published results, however, the preparation of the Alliaceae extract was not described, I ask that this description be made in detail.
we apologize for the omission, we acquired a concentrated liquid commercial Alliaceae extract, Garlicon (DOMCA SAU, Granada, Spain), it has been used by other researchers with positive effects on birds productivity, we encapsulate it into a dextrin-lecithin matrix and validated by the presence of PTSO which has a concentration of 12 g.kg-1, description is in Lines 137 to 141
Display equipment information.
Gas chromatograph model 7890A Agilent Technologies Inc, coupled to a simple quadrupole mass detector model 5975C Agilent Technologies Inc., Santa Clara, CA, USA.
Table 2 must be cited before Table 3.
The tables below provide comprehensive data on the interactions between Eimeria challenged and AE-e supplementation (Table 2), and subsequently present independently the effects of Eimeria challenge (Table 3), and the effects of AE-e supplementation (Table 4) on ADG, ADFI, and FCR.
For table 3, it is important to highlight that in the finisher phase there were no variations and to discuss the variations in the cumulative study.
Despite not finding any changes in the finisher phase (p>0.05), the negative effect observed in the starter phase continued to be observed in the cumulative period, ADG (p<0.001) and FCR (p=0.01) in broiler chickens from C group compared to NC group. However, the ADFI was not affected by the challenge (p>0.05). (Table 3).
Indicate according to Figures 3b, 3d and 3f.
We include it as you recommended, Done Line 501
This part of the discussion can be removed from the text as it was already presented in the introduction of the work.
We agree, we eliminated from the article
Specify the species, A. sativum or A. cepa?
Similar results were found by Elkhtam, et al. [67] where OPG, the clinical symptoms of the disease and the LS decreased with the addition of garlic extract (A. sativum) to the diet of broiler chickens challenged with Eimeria spp.
I did not find reference number 68 in the text.
Reference 68 Ali et al. 2019 (now 65) Is located in the previous paragraph, line 568.
A similar effect was observed on OPG of broiler challenged with Eimeria and supplemented with garlic extracts (A.sativum) [64,65]
I suggest looking for a more current reference.
Done, now is 2023. Line 619
References need to be carefully reviewed as they do not comply with the journal's standards. Scientific names are not in italics, some titles are written in capital letters, some journal names are abbreviated and others are not. According to journal standards, journal names must be abbreviated.
Thank you for your advice, now references are carefully written following Journal standards. You are not going to find all your suggestions because we substitute several older references for some current references. We use Italics in latin names as Allium and Eimeria.
Reviewer 2 Report
Comments and Suggestions for Authors
Author Response
Dear reviewer, we want to thank you for the time and dedication you have given to the review of our manuscript, undoubtedly your comments have been very useful to make a better proposal both in form and in substance. In the next paragraph you will find the response to your comments and suggestions. Attached you will find the corrected version of our paper. We hope that the corrections and explanations we have annexed meet your expectation.
RESPONSE TO COMMENTS AND SUGGESTIONS
REVIEWER 2
Title: Effect of Alliaceae encapsulated extract on growth performance, gut health, and intestinal
microbiota in broiler chickens challenged with Eimeria spp.
General comments: The manuscript is well written. Moreover, the study's experimental design and data analysis are scientifically structured. The findings in this study also provide additional critical information that relates the effects of Phytochemicals such as "Alliaceae" to alleviate coccidiosis in broiler chickens and its effects on the intestinal microbiota of infected, infected-treated chickens. Some points and clarifications are listed below for the authors to consider in their revision prior to acceptance for publication.
1.) What about adding and expressing a cumulative finding on the effects of Alliaceae
encapsulated extract following ACI or Anticoccidial Index?
Your observation adds a lot of value to our manuscript. You will find the incorporation of the ACI concept in the following tirets:
-Material and methods –
2.9 Anticoccidial index (ACI), lines from 207 to 222
2.13 Statistical Analyses, lines 272-273 and 277-279
Results –
3.4 Anticoccidial index (ACI), from lines 407 to 420
Discussion- Lines 569-571.
Conclusion- Lines 659-660
2.) The authors were able to correlate the performance and gut health following the production parameters (ADG, FCR), OPG, and LS. While all information collected is vital, a histopathological section on selected intestinal sections is similarly essential to see the effects of treatments on the intestinal villi and crypts for a holistic representation of the effects of the treatment in the gut. What about histopathological sections of intestines in this study?
Your observation is valuable, unfortunately, in this study we did not make the histological measurements that you correctly suggest. We have recently concluded a study in the same topic in which we made the measurements of villus length, crypt depth and the estimation of the V/C.
3.) In Table 5, OPG was presented separately as Total OPG, then E. acervulina, E. maxima, and E. tenella. Based on the materials and methods section, the infection was given, in L165, "...with 0.5 mL of a mixture of sporulated oocysts". Why was OPG presented also as an individual count per species, and how was this carried out?
An expert on this topic differentiated the Eimeria sp. by the morphological characteristics of sporulated oocysts, the number of oocysts was expressed as OPG. The total OPG is the sum of OPG for all three species, E. acervulina, E. maxima, and E. tenella. Description is added on Lines 196-198.
- Materials and Methods Section
2.2 Housing, Animals and Experimental Design Please include the ionophores used and the concentration in the section (L120).
Example: following a program of 50ppm nicarbazin-60ppm salinomycin; 50ppm narasin-60ppm salinomycin
The description of the anticoccidial program is in Lines 161-162.
2.3 Alliaceae encapsulated extract supplementation
As the primary agent used in the study, please include the source and purity of the Alliaceae extract used, including plant preparation/extraction and analysis.
We apologize for the omission, we acquired a commercial concentrated Alliaceae liquid extract, Garlicon, (DOMCA SAU, Granada, Spain), which has been used by other researchers with positive effects on birds performance. We encapsulate it into a dextrin-lecithin matrix and validated by the presence of PTSO which has a concentration of 12 g.kg-1, description is in Lines 137 to 141
2.8 Intestinal Lesion Score (LS)
In the results section, LS was also calculated and expressed as TMLS. Thus, authors are advised to include and mention TMLS also in this section.
Description of TMLS is added in Line 220
- Other corrections (spelling, format etc.)
Throughout the manuscript, be consistent with the spelling and format for the following:
- Species is always spelled "species" (singular or plural). Change 'specie' to "species".
Example: Introduction, L55 '...,each specie causes' to "...,each species causes"
We appreciate your advice, “species” is written 10 times, and now all of them are “species” no matter if is singular or plural.
- Change 'E. acervuline' and 'E Acervulina' to "E. acervulina"
2 Example: In lines L57, 315, L325, 355, Table 5,
Sorry for the inconsistency, now “E. acervulina” appears in 27 occasions and is written as you recommended.
- Italics or not. However, be consistent in the use and format for spp. throughout
the manuscript. Done, now are not Italic.
- In L65, change 'To the date, the coccidiosis...' to " To date, coccidiosis..." Done, Line 72
- In L66, 'decoquinato' to "decoquinate" Done, Line 73
- In L68, change '...diet or in drinker water' to "...diet or drinking water" Done, Line 74
- In L72, delete 'the' in "the coccidiosis"..."the phytochemicals" Done, Line 78 and 79
- In L146, change 'started' to "starter" Done, Line 162
- In L285, change '...was no affected' to "...was not affected" Done, Line 325
- In L455, change 'Ours result show...' to "Results from this study show..." Done, Line 505
Reviewer 3 Report
Comments and Suggestions for Authors
This manuscript is well written and format. I only several minor suggestions regarding the writing:
Line:
61: Eimeria spp. Spp. should not be italicized, and there must be a “.” after spp.
the abbreviation "sp." or "spp." used following the genus name, meaning species or species plural respectively. Please note that the "sp."/"spp." abbreviations are never capitalized or italicized, although the preceding genus name is. A full stop/period is always placed after these abbreviations unless used at the end of a sentence and would result in a double full stop/period. Please revise them accordingly in the following sections too.
68 “[12],” then please add a space after that
115: keeping between 18-21 C? Please revise “a”
Section 2.2- Housing and experimental design: Is this management based on the cobb-500 management guide? If so, please add that information. 1 hrs of dark seems a little bit too short for the growth stage, and cobb required longer dark hours around the 21 days old age (Start increasing the dark period when the birds reach 130 to 180 g). Could you please explain that?
229: space after " ]"
376: please remove the space after (
573: spp. should not be italicized
Comments on the Quality of English LanguageThis manuscript is well-written and grammatically correct.
Author Response
We are very grateful to you for your comments and the time you have spent reviewing our article. Attached you will find the file that we have edited with the comments of the reviewers, which have enriched the writing.
RESPONSE TO COMMENTS AND SUGGESTIONS
REVIEWER 3
Comments and Suggestions for Authors
This manuscript is well written and format. I only several minor suggestions regarding the writing:
Line:
61: Eimeria spp. Spp. should not be italicized, and there must be a “.” after spp.
the abbreviation "sp." or "spp." used following the genus name, meaning species or species plural respectively. Please note that the "sp."/"spp." abbreviations are never capitalized or italicized, although the preceding genus name is. A full stop/period is always placed after these abbreviations unless used at the end of a sentence and would result in a double full stop/period. Please revise them accordingly in the following sections too.
Your observation is much appreciated. We made corrections to the ones that were written incorrectly. Now there are 35 “spp.” correctly written in the manuscript.
68 “[12],” then please add a space after that
The correction is in line 75, in addition we move the reference forward.
115: keeping between 18-21 C? Please revise “a”
Done, now in Line 127
Section 2.2- Housing and experimental design: Is this management based on the cobb-500 management guide? If so, please add that information. 1 hrs of dark seems a little bit too short for the growth stage, and cobb required longer dark hours around the 21 days old age (Start increasing the dark period when the birds reach 130 to 180 g). Could you please explain that?
As for the point in which you ask us the clarification of the times of darkness/ lighting in the reception of the chick, in fact we have based on the recommendations of the genetic Cobb 500 line in our conditions of production, in which the first 4 days we offer 23 hours of light so that the chicks can find access to feeder and drinker in addition to locating their new space in the pen and socialize with their companions. When they reach a weight of around 100 grams on the fifth day of age, we apply the natural photoperiod, which is characterized by 9 to 10 hours of darkness in Querétaro, México.

229: space after " ]"
Done, now in Line 266.
376: please remove the space after (
Done, now in Line 430.
573: spp. should not be italicized
Done, now in Line 618.
Comments on the Quality of English Language
This manuscript is well-written and grammatically correct.
Submission Date
25 October 2023
Date of this review
15 Nov 2023 17:21:34
Round 2
Reviewer 1 Report
Comments and Suggestions for Authors
The requested changes were made by the authors, thus the article was considered accepted.
Author Response
Thank you so much for your suggestions and comments related to our article.